# CAN DEEPFAKE SPEECH BE RELIABLY DETECTED?

## ABSTRACT

Recent advances in text-to-speech (TTS) systems, particularly those with voice cloning capabilities, have made voice impersonation readily accessible, raising ethical and legal concerns due to potential misuse for malicious activities like misinformation campaigns and fraud. While synthetic speech detectors (SSDs) exist to combat this, they are vulnerable to "test domain shift", exhibiting decreased performance when audio is altered through transcoding, playback, or background noise. This vulnerability is further exacerbated by deliberate manipulation of synthetic speech aimed at deceiving detectors. This work presents the first systematic study of such active malicious attacks against state-of-the-art open-source SSDs. White-box attacks, black-box attacks, and their transferability are studied from both attack effectiveness and stealthiness, using both hardcoded metrics and human ratings. The results highlight the urgent need for more robust detection methods in the face of evolving adversarial threats.

## 1 INTRODUCTION

Recent years have witnessed a remarkable advance in text-to-speech (TTS) systems (Kreuk et al., 2022; Borsos et al., 2023; Leng et al., 2023; Saeki et al., 2023; Shen et al., 2023; Wang et al., 2023; ChatTTS, 2024; Chen et al., 2024; xTTS, 2024; ElevenLabs, 2024; Le et al., 2024; Lux et al., 2024). Some of these systems possess the zero-shot / few-shot voice cloning capability (Biadsy et al., 2024; Cooper et al., 2020; Casanova et al., 2022; Ye et al., 2024; Le et al., 2024), that can mimic someone's voice using only a brief sample of that person's speech recording. The widespread access of these systems, either through open-source projects or commercial APIs, has made voice impersonation easier than ever. This raises significant ethical and legal concerns as the capability can be easily misused for misinformation campaign, fraud, copyright infringement, *etc*. As an instance, a scammer utilized synthetic audio to mimic President Biden in unlawful robocalls during a New Hampshire primary election, resulting in a \$6 million penalty and felony accusations (Coldewey, 2024). Besides, numerous instances of synthetic audio misuses can be found online if searching with "deepfake audio", which underscores the urgent need to address this growing problem.

Synthetic speech detectors (SSDs) are deployed to mitigate the misuse of synthetic speech. While there are a number of advanced SSDs (Tak et al., 2021c;a; Jung et al., 2022), recent research (Müller et al., 2022; Xie et al., 2024) suggests they might struggle when facing "test domain shift". Concretely, a detector may exhibit decreased performance when presented with audio that has undergone alterations, including transcoding, playback, background noise, or even just a shift in the TTS system used. However, current research efforts are directed towards these natural changes to audio, it is important to recognize that deliberate manipulation of synthetic speech by an attacker with the intent to deceive detectors can significantly increase the likelihood of success of the attacker. However, there is a lack of systematic research on these malicious perturbations.

In this work, we conduct the first systematic study of active malicious attacks against the state-of-the-art open-source SSDs. We investigate a range of attack scenarios, considering adversaries with varying levels of access to the target SSD: those with full knowledge of the model (white-box), those who can only interact with it and observe the results (black-box), and those who cannot even query the model (agnostic); and evaluate the results using both hard-coded metrics and human ratings.

Our findings reveal that:

---
*Equal Contribution

- Increased access to the detector makes it easier for attackers to create deepfakes that can evade detection without any noticeable loss in audio quality.

- Existing open-source SSD detectors are vulnerable when facing synthetic audio generated by TTS systems never seen during training.

- VisQOL scores and human ratings show that the audio quality after attack is reasonable.

- Alarmingly, even in the agnostic setting, attackers can still bypass state-of-the-art open-source SSDs with reasonable chance.

**Overall, we need more robust SSDs to mitigate the growing threat of deepfake audio misuse.**

## 2 PRELIMINARY

### 2.1 TTS TECHNIQUES

TTS systems, which convert written text into speech, have a long history of development and have made remarkable progress in recent years. Early TTS systems primarily used a concatenative approach (Khan & Chitode, 2016), where speech was synthesized by joining pre-recorded speech units from a database. Despite the simplicity, they suffered from unnatural prosody and robotic-sounding speech. To address the issue, researchers proposed statistical parametric speech synthesis (Zen et al., 2009). These systems used statistical models to learn the relationship between linguistic features (e.g., phonemes, part-of-speech tags) and acoustic features (e.g., fundamental frequency, spectral envelope) and enabled more natural-sounding speech generation with improved prosody.

Recently, deep learning has revolutionized the field of TTS. Neural network-based TTS systems have surpassed traditional methods in terms of speech naturalness and intelligibility. Several architectures have been explored, including sequence-to-sequence models (Wang et al., 2017; Ping et al., 2017), attention-based models (Ren et al., 2019; 2020), and generative adversarial networks (GANs) (Kumar et al., 2019). These models can directly learn the mapping from text to speech, enabling end-to-end training and eliminating the need for complex feature engineering. Some of the popular neural TTS systems include Kreuk et al. (2022); Borsos et al. (2023); Leng et al. (2023); Saeki et al. (2023); Shen et al. (2023); Wang et al. (2023); ChatTTS (2024); Chen et al. (2024); xTTS (2024); ElevenLabs (2024); Le et al. (2024); Lux et al. (2024).

### 2.2 SYNTHETIC SPEECH DETECTION TECHNIQUES

In the past, detecting synthetic speech required carefully crafted features (Doddington et al., 2001; Alegre et al., 2013; Hanilçi et al., 2015; Patel & Patil, 2015; Sahidullah et al., 2015; Todisco et al., 2016). However, with the increase in available data and the development of larger models, simpler features like waveforms or spectrograms are now sufficient for effective detection. RawNet2 (Tak et al., 2021c) is a deep convolutional neural network for synthetic speech detection using merely raw waveforms. It builds upon the RawNet (Jung et al., 2020) architecture by incorporating residual connections and dilated convolutions. RawNetGATST (Tak et al., 2021a) extends RawNet2 by incorporating a graph attention network (Tak et al., 2021b) to identify key spectral or temporal features for detection. Similarly, AASIST (Jung et al., 2022) refines the graph network architecture further for improved synthetic speech detection.

## 3 BYPASS SYNTHETIC SPEECH DETECTION SYSTEMS

In this section, we aim to answer the following question:

**Can deepfake audio be altered in ways nearly imperceptible to the human ear, but sufficient to bypass state-of-the-art detectors?**

Unlike previous research that focused on natural perturbations (Müller et al., 2022; Xie et al., 2024), we consider a malicious attacker who deliberately optimizes the perturbation to evade detection. We examine this scenario under various levels of access to the detection systems, from having full knowledge (white-box), to partial knowledge (black-box), to no knowledge (agnostic).

Table 1: Baseline EERs of SSDs on the ASVSpoof2019-LA test split without attacks.

| AASIST | AASIST-L | RawNet2 | RawGATST |
|--------|----------|---------|----------|
| 0.83% | 0.99% | 4.88% | 3.29% |

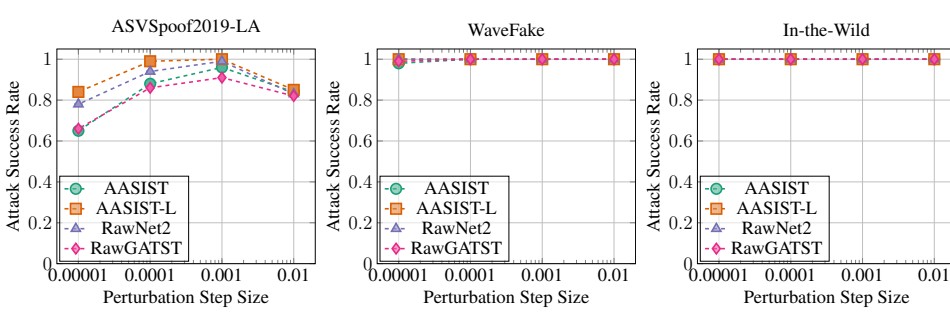

(a) Attack success rate vs. perturbation step size in PGD.

(b) ViSQOL score vs. perturbation step size in PGD.

Figure 1: Attack success rate and ViSQOL vs. perturbation step size in PGD.

## 3.1 EXPERIMENT SETUP

We train four SOTA open-source SSDs from scratch: AASIST (Jung et al., 2022), AASIST-L (Jung et al., 2022), RawNet2 (Tak et al., 2021c) and RawGATST (Tak et al., 2021a) on ASVSpoof2019-LA train split (Todisco et al., 2019). Their equal error rates (EERs) without attacks on ASVSpoof2019-LA test split are reported in Table 1, and closely match the reported numbers in their original papers.

We launch attacks on three synthetic datasets: ASVSpoof2019-LA test split (Todisco et al., 2019), WaveFake (Frank & Schönherr, 2021) and In-the-wild (Müller et al., 2022). In consideration of compute resources, we randomly sub-sample 100 examples from each dataset for the attacks.

We use the attack success rate (*i.e.* the ratio of attacked examples bypassing the target detector) to measure the effectiveness of the attacks. To ensure the attack does not degrade audio quality, we use both VisQOL (Hines et al., 2015) and human ratings to confirm that the attacked audio still sounds similar to the original synthetic audio, which we refer to as "stealthiness".

## 3.2 WHITE-BOX ATTACK

**Takeaway:**
- Existing open-source SSDs are extremely vulnerable to white-box attacks.
- Deepfake speech from TTS not seen during training is more likely to bypass SSDs.
- White-box attacks can be highly effective and stealthy simultaneously.

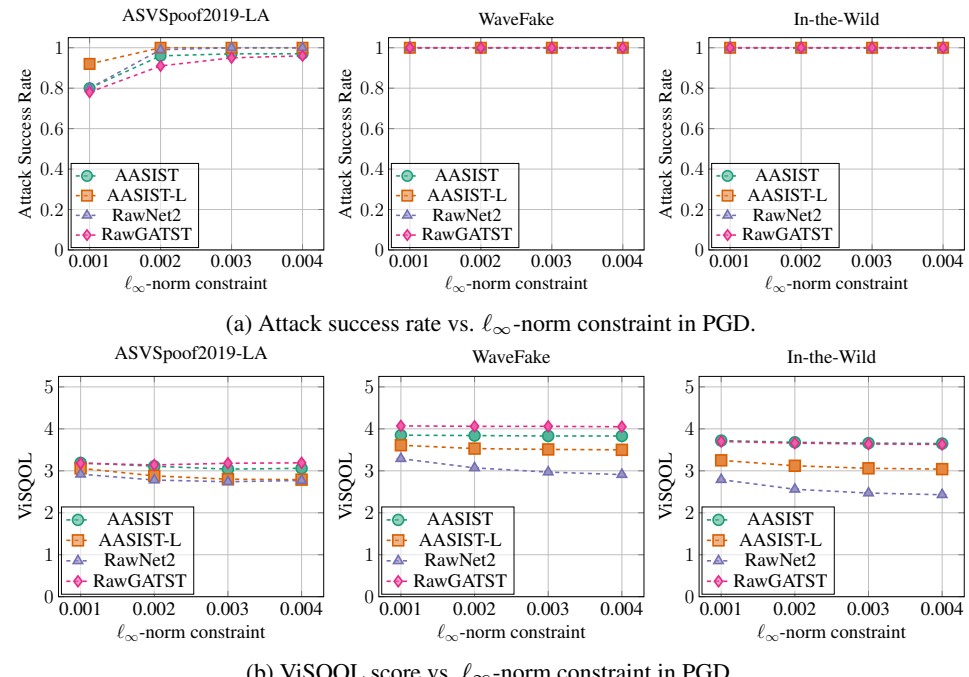

(a) Attack success rate vs. $\ell_\infty$-norm constraint in PGD.

(b) ViSQOL score vs. $\ell_\infty$-norm constraint in PGD.

Figure 2: Attack success rate and ViSQOL vs. $\ell_\infty$-norm constraint in PGD.

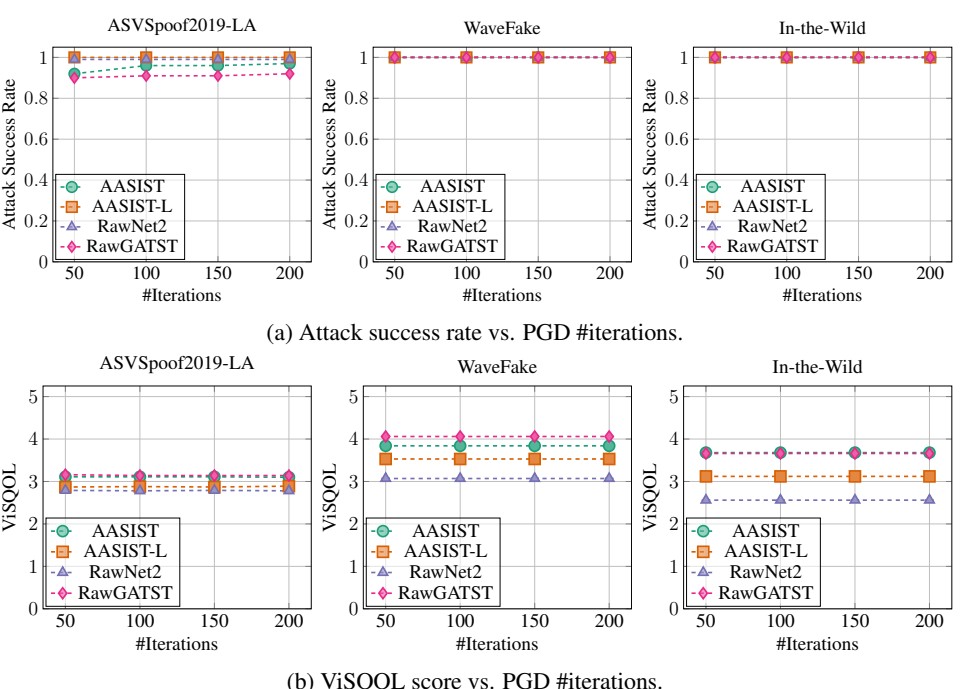

(a) Attack success rate vs. PGD #iterations.

(b) ViSQOL score vs. PGD #iterations.

Figure 3: Attack success rate and ViSQOL vs. PGD #iterations across red-team sets.

We first study white-box attack, where the adversary has full access to the model. We choose two white-box attacks: Projected Gradient Descent (PGD) (Mkadry et al., 2017) and I-FGSM (Kurakin et al., 2018). Algorithm 1 in Appendix shows details of PGD and I-FGSM.

**Projected Gradient Descent:** PGD crafts adversarial examples by iteratively taking small steps in the direction that maximizes the model's error, while projecting the perturbed example back within

Table 2: Human ratings of speaker similarity between the original and PGD attacked audio.

|  | ASVspoof | WaveFake | In-the-wild |
|---|---|---|---|
| AASIST | $0.970 \pm 0.063$ | $0.971 \pm 0.046$ | $0.985 \pm 0.030$ |
| AASIST-L | $0.979 \pm 0.037$ | $0.979 \pm 0.045$ | $0.975 \pm 0.036$ |
| RawNet2 | $0.971 \pm 0.077$ | $1.000 \pm 0.000$ | $0.967 \pm 0.063$ |
| RawGATST | $0.997 \pm 0.008$ | $0.986 \pm 0.030$ | $0.997 \pm 0.008$ |

a certain boundary around the original input to maintain a balance between attack success rate and stealthiness.

PGD has three major hyper-parameters: perturbation step size, $\ell_\infty$-norm constraint, and the number of iterations. We conduct hyper-parameter search and summarize the results in Figure 1, 2, and 3.

In Figure 1, we can tell that on WaveFake and In-the-wild, the attack success rate is almost always 100% while on ASVSpoof2019-LA test the attack success rate hovers between 60% and 100% depending on the learning rate used. This reflects the fact that the detectors are more robust on test data generated by the same TTS systems as the training data (*i.e.* in-domain data), but are still vulnerable under white-box attacks with a few steps of hyper-parameter search. On the other hand, VisQOL scores keep decreasing as the perturbation step size grows. Usually, VisQOL score above 3.0 is considered reasonable quality. Thus, there exists a sweet spot of perturbation step size striking balance between attack effectiveness and stealthiness.

In Figure 2, the observation of SSDs being more robust on ASVSpoof2019-LA test holds true. However, we observe that the VisQOL scores are pretty consistent despite the changing $\ell_\infty$-norm constraint, which says that audio quality is insensitive to $\ell_\infty$-norm constraint within a certain range.

Figure 3 shows that white-box attacks are efficient, reaching maximum attack success rates and stable VisQOL scores after just 50 iterations.

We also collect human ratings on whether the PGD-attacked audio with the best hyper-parameter combination sounds like the original synthetic audio, and the results are summarized in Table 2. We can see that most human raters think the two audio sound like the same person, underscoring the potential threat of using the attacked audio for impersonation.

**Iterative Fast Gradient Sign Method:** I-FGSM only differs from PGD in that it only uses the sign of the gradient to perturb the input audio. It shares the same set of hyper-parameters as PGD, for which the grid search results are summarized in Figure 4, 5, and 6 and human ratings are summarized in Table 3, and the findings are similar to PGD.

### 3.3 BLACK-BOX ATTACK

**Takeaway:**
- Existing open-source SSDs are still vulnerable to black-box attacks.
- Black-box attacks can be effective and stealthy simultaneously.

For black-box attack, we choose the Simple Black Box Attack (SimBA) (Guo et al., 2019). SimBA perturbs the input audio randomly and observes whether the prediction confidence score for "fake" class decreases or increases. If the confidence score decreases, SimBA will keep the perturbation. Otherwise, SimBA will try adding perturbation in the opposite direction and decide whether to

Table 3: Human ratings of speaker similarity between the original and I-FGSM attacked audio.

|  | ASVspoof | WaveFake | In-the-wild |
|---|---|---|---|
| AASIST | $0.984 \pm 0.020$ | $0.960 \pm 0.052$ | $0.985 \pm 0.024$ |
| AASIST-L | $0.987 \pm 0.022$ | $0.986 \pm 0.023$ | $0.967 \pm 0.054$ |
| RawNet2 | $0.980 \pm 0.040$ | $1.000 \pm 0.000$ | $0.991 \pm 0.012$ |
| RawGATST | $0.989 \pm 0.024$ | $0.858 \pm 0.141$ | $0.985 \pm 0.024$ |

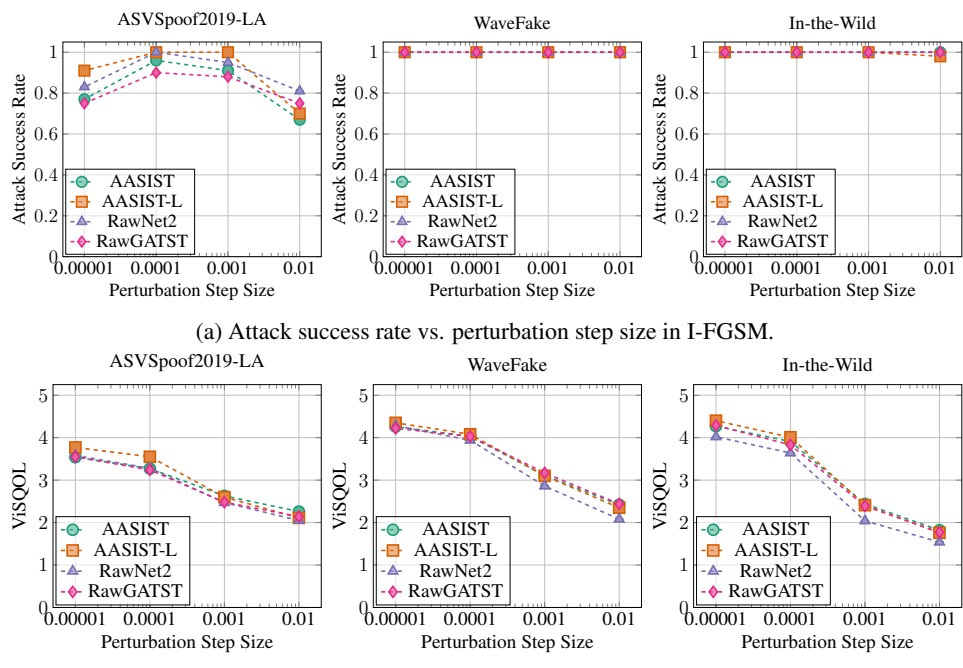

(a) Attack success rate vs. perturbation step size in I-FGSM.

(b) ViSQOL score vs. perturbation step size in I-FGSM.

Figure 4: Attack success rate and ViSQOL vs. perturbation step size in I-FGSM.

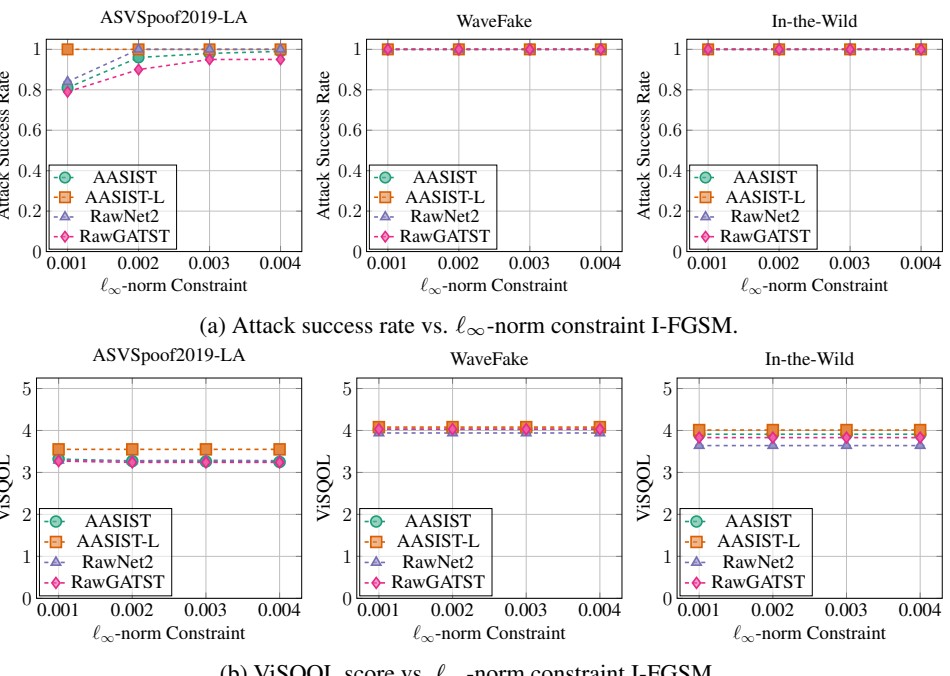

(a) Attack success rate vs. $\ell_\infty$-norm constraint I-FGSM.

(b) ViSQOL score vs. $\ell_\infty$-norm constraint I-FGSM.

Figure 5: Attack success rate and ViSQOL vs. $\ell_\infty$-norm constraint in I-FGSM.

keep or discard the perturbation just as above. SimBA iteratively perturbs the input until the SSD is successfully bypassed or the budget of queries/iterations is used up. Algorithm 2 in Appendix shows the details of SimBA.

SimBA has three hyper-parameters: perturbation batch size, perturbation step size, and the number of queries. Perturbation batch size decides how many timesteps are perturbed in each query, while

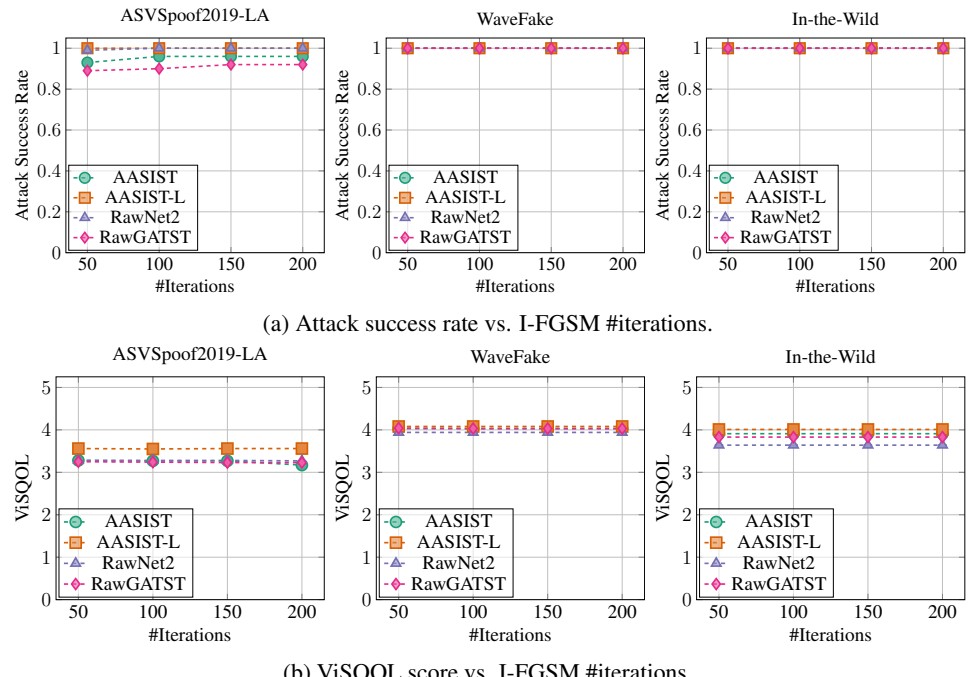

(a) Attack success rate vs. I-FGSM #iterations.

(b) ViSQOL score vs. I-FGSM #iterations.

Figure 6: Attack success rate and ViSQOL vs. I-FGSM #iterations.

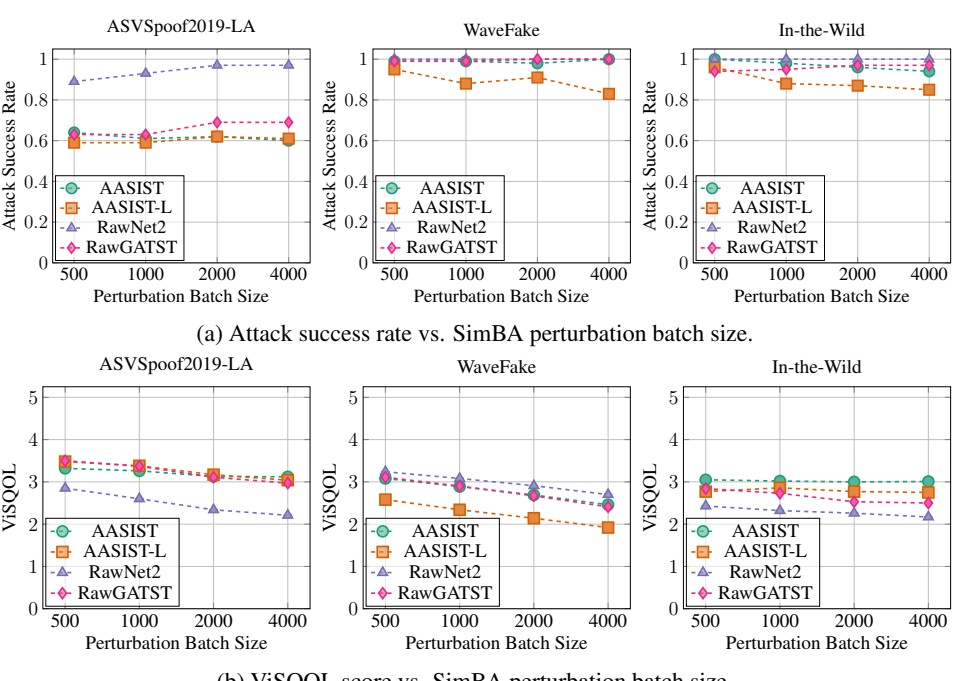

(a) Attack success rate vs. SimBA perturbation batch size.

(b) ViSQOL score vs. SimBA perturbation batch size.

Figure 7: Attack success rate and ViSQOL vs. SimBA perturbation batch size.

perturbation step size decides which long one perturbation step on one timestep can be. The hyper-parameter search results are summarized in Figure 7, 8, and 9.

In Figure 7, we observe that on ASVSpoof2019 test, RawNet2, the least capable SSD model is still broken almost 100% but all the other 3 models are only broken 60% of all the tested examples. This draws a positive correlation between model capability and robustness. On WaveFake and In-the-

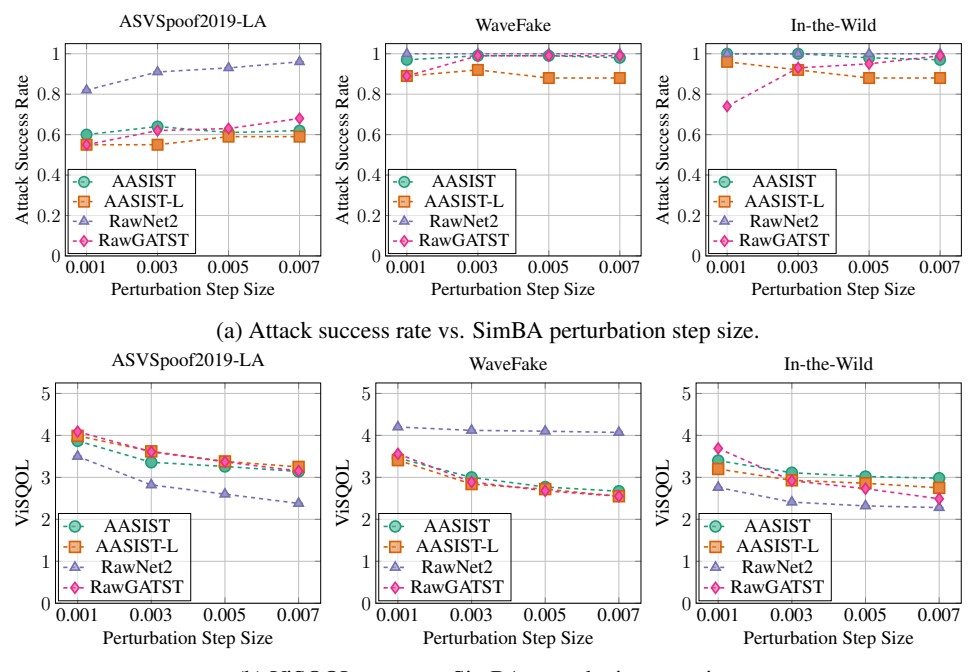

(a) Attack success rate vs. SimBA perturbation step size.

(b) ViSQOL score vs. SimBA perturbation step size.

Figure 8: Attack success rate and ViSQOL vs. SimBA perturbation step size across datasets.

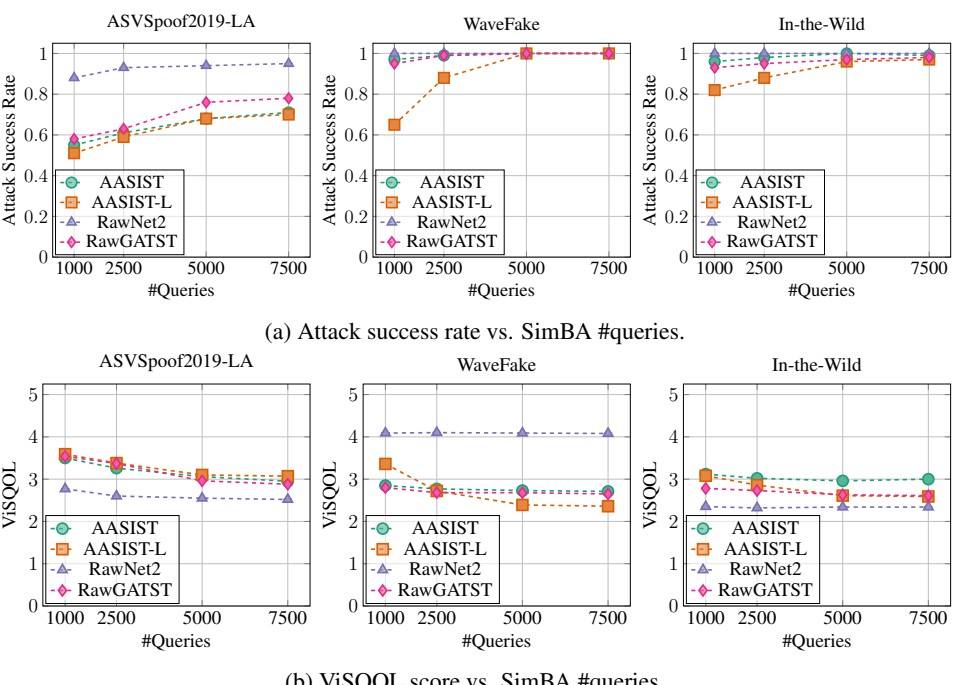

(a) Attack success rate vs. SimBA #queries.

(b) ViSQOL score vs. SimBA #queries.

Figure 9: Attack success rate and ViSQOL vs. SimBA #queries across datasets.

Wild, all SSDs are broken more than 90% of the time, which confirms the previous observation that current SSD models are brittle when facing synthetic audio from TTS systems never seen during training.

Table 4: Human ratings of speaker similarity between the original and simBA attacked audio.

|  | ASVspoof | WaveFake | In-the-wild |
|---|---|---|---|
| AASIST | $0.984 \pm 0.020$ | $0.960 \pm 0.052$ | $0.985 \pm 0.024$ |
| AASIST-L | $0.987 \pm 0.022$ | $0.986 \pm 0.023$ | $0.967 \pm 0.054$ |
| RawNet2 | $0.980 \pm 0.040$ | $1.000 \pm 0.000$ | $0.991 \pm 0.012$ |
| RawGATST | $0.989 \pm 0.024$ | $0.858 \pm 0.141$ | $0.985 \pm 0.024$ |

Also in Figure 7, 8, and 9, we observe that ASSIST-L is the most robust model consistently, which is surprising because it's the smallest model within the 4 (See Jung et al. (2022) for the size of these models.). This observation aligns with the principle of Occam's razor, which suggests that simpler models often generalize better. A potential explanation could lie in the raggedness of the decision boundaries. Larger models, with their increased complexity, might create more intricate and potentially overfit decision boundaries. In contrast, ASSIST-L, being smaller, may form smoother decision boundaries, leading to better generalization and robustness against perturbations.

Human ratings of audio similarity is summarized in Table 4. Again the attacked audio sound highly similar to the original ones to human ears.

## 3.4 AGNOSTIC ATTACK: TRANSFERABILITY OF ABOVE ATTACKS

> **Takeaway:**
> - For both white-box attacks and black-box attacks, transferability depends on the target model's capability on the target audio.
> - Black-box attacks are more transferrable on in-domain test data than out-of-domain data.
> - Transferrability of different white-box attacks are alike.

The above attacks all assume different levels of access to the SSD model which might not be accessible in practice. As a result, we want to understand whether the above attacks are transferrable: Can a successfully attacked example on one model transfer to a different model? If this is true, then the adversary can craft a proxy model themselves, attack it, and expect it to bypass the real SSD as well.

The results are summarized in Figure 10. First, we find that on out-of-domain data, some SSDs are extremely vulnerable. For example, on WaveFake, RawNet2 is extremely vulnerable under all attacks; on In-the-wild, ASSIST and AASIST-L are more vulnerable than the other two models. Second, we find that on in-domain data, black-box attacks are much more transferrable than white-box attacks. This is because 1) black-box attacks tend to add larger perturbation than white-box attacks; 2) the SSDs' decision borders are alike for in-domain data. Thirdly, we also observe high similarity between the transferrability heatmap between PGD and I-FGSM, which might be due to different white-box attacks taking gradient paths in similar directions despite small differences.

## 4 CONCLUSION

This work presents the first systematic exploration of the robustness of state-of-the-art SSDs against adversarial attacks. Our findings reveal critical implications for SSD deployment and future research.

Firstly, we demonstrate a clear correlation between system accessibility and vulnerability. Open-access SSDs, and even those with oracle access, are highly susceptible to attacks. This underscores the critical need to restrict public access to SSD models and internal workings. While complete prevention of information leakage may be challenging, measures such as rate limiting can effectively mitigate the threat of black-box attacks.

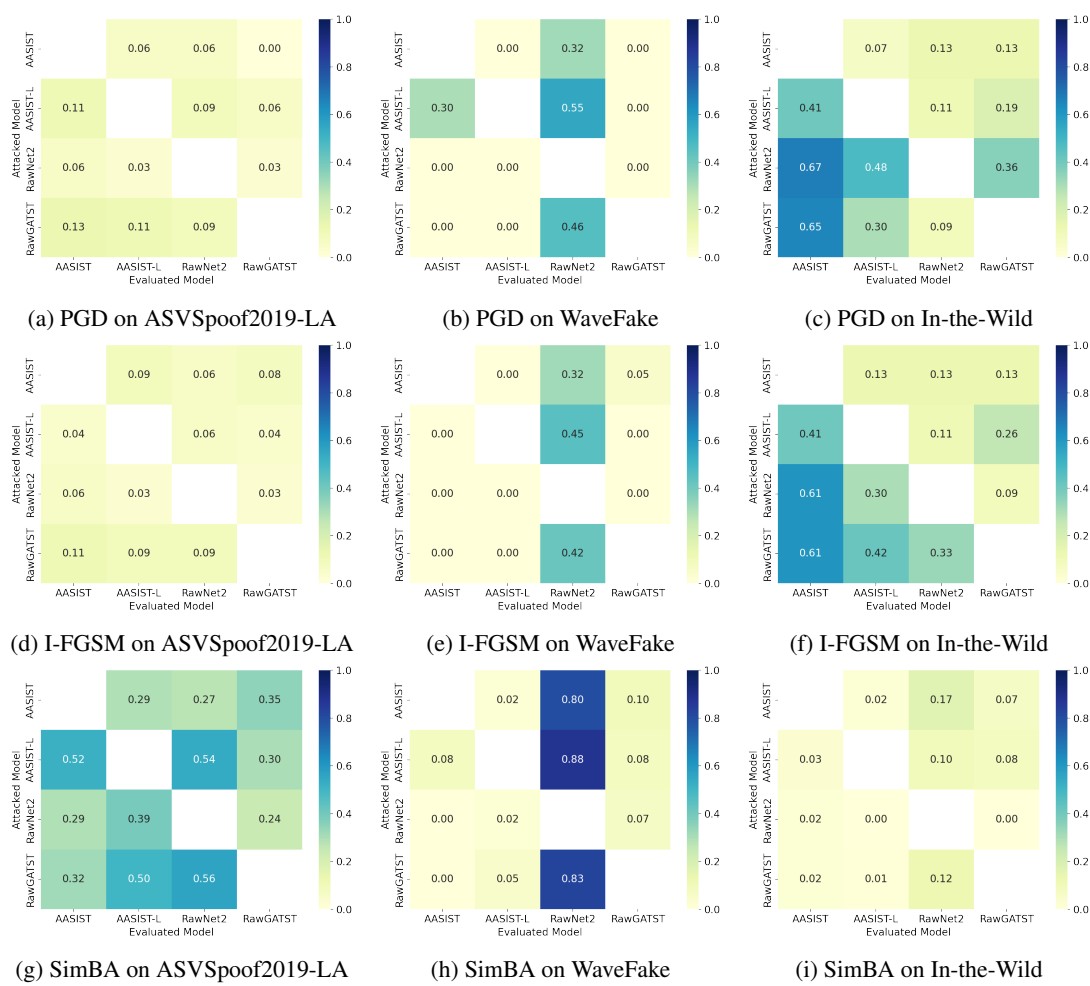

Figure 10: Transferability of attacks on different datasets.

Secondly, our research exposes the vulnerability of current open-source SSDs to domain shifts, including active perturbations and, notably, synthetic data generated from unseen TTS systems. This highlights two key recommendations:

- Comprehensive and Continuously Updated Training Data: SSD training datasets should encompass a diverse range of TTS systems and be regularly updated to incorporate new systems, ensuring broad coverage against evolving spoofing techniques.

- Data Composition Confidentiality: Maintaining the confidentiality of training data composition is crucial to prevent attackers from exploiting this knowledge for targeted attacks, as evidenced by the significant advantage gained from TTS system shifts even in agnostic attacks.

Our findings also draw a parallel between the interplay of SSD and TTS development and an offline GAN, suggesting a future where TTS systems may achieve near-perfect human mimicry. This emphasizes the crucial need for complementary audio authenticity techniques, such as robust watermarking (Liu et al., 2024), to bolster the security of SSD systems.

In conclusion, this study serves as a critical analysis of the current state of open-source SSD robustness. By exposing key vulnerabilities and providing actionable recommendations, we aim to guide future research and development efforts towards building more secure and resilient SSD systems in an ever-evolving landscape of speech synthesis and spoofing technologies.

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

---

**Algorithm 1** White-box Attacks: PGD and I-FGSM

---

**Input:** Waveform audio $s \in \mathbb{R}^T$, SSD model $f$, perturbation step size $\alpha$, maximum number of iterations $T$, real audio class $\mathcal{R}$, attack $A \in \{$'PGD', 'I-FGSM'$\}$, $\ell_\infty$-norm constraint $\epsilon$
**Output:** Adversarial perturbation $\delta \in \mathbb{R}^T$
1: $\delta \leftarrow 0$ ▷ initialization
2: **for** $t \leftarrow 1$ to $T$ **do**
3:    **if** $A ==$ 'PGD' **then**
4:       $\delta \leftarrow \delta - \alpha \cdot \nabla_\delta l_{CE}(f(s+\delta), \mathcal{R})$    ▷ gradient descent with cross entropy loss $l_{CE}$
5:    **end if**
6:    **if** $A ==$ 'I-FGSM' **then**
7:       $\delta \leftarrow \delta - \alpha \cdot \text{sign}(\nabla_\delta l_{CE}(f(s+\delta), \mathcal{R}))$    ▷ FGSM with cross entropy loss $l_{CE}$
8:    **end if**
9:    $\delta \leftarrow \text{clip}(\delta, -\epsilon, \epsilon)$ ▷ projection
10:   **if** $f(s+\delta) == \mathcal{R}$ **then** ▷ attack success
11:       **Break**
12:   **end if**
13: **end for**
14: **return** $\delta$

---

**Algorithm 2** Black-box Attack: SimBA

---

**Input:** Waveform audio $s \in \mathbb{R}^T$, SSD model $f$, perturbation step size $\alpha$, perturbation batch size $q$, maximum number of queries $Q$, real audio class $\mathcal{R}$, $\ell_\infty$-norm constraint $\epsilon$
**Output:** Adversarial perturbation $\delta \in \mathbb{R}^T$
1: $\delta \leftarrow 0, t \leftarrow 0$ ▷ initialization
2: $p \leftarrow f(s, \mathcal{R}), t \leftarrow t + 1$    ▷ initializing highest probability to predict real audio class
3: **while** $t < T$ **do**
4:    **if** $f(s+\delta) == \mathcal{R}$ **then** ▷ attack success
5:       **Break**
6:    **end if**
7:    $r \in \mathbb{R}^T$ and $r \leftarrow 0$
8:    Randomly choose $q$ dimensions from $r$ without replacement
9:    Randomly add $\alpha$ or $-\alpha$ to the chosen $q$ dimensions in $r$
10:   $t \leftarrow t + 1$ ▷ one more query for $f$ below
11:   **if** $f(s+\delta+r, \mathcal{R}) > p$ **then**
12:       $p \leftarrow f(s+\delta+r, \mathcal{R})$    ▷ update highest probability to predict real audio class
13:       **Continue**
14:   **else**
15:       $t \leftarrow t + 1$ ▷ one more query for $f$ below
16:       **if** $f(s+\delta-r, \mathcal{R}) > p$ **then**
17:          $p \leftarrow f(s+\delta-r, \mathcal{R})$    ▷ update highest probability to predict real audio class
18:          **Continue**
19:       **end if**
20:   **end if**
21: **end while**
22: **return** $\delta$

---

