# OpenReview forum: "CAN DEEPFAKE SPEECH BE RELIABLY DETECTED?"
_ICLR.cc/2025/Conference — ICLR 2025 Conference Withdrawn Submission_

### Official Review · Reviewer_AHV9 · 2024-10-29

**Soundness:** 2
**Presentation:** 2
**Contribution:** 1
**Rating:** 3
**Confidence:** 5

**Summary:**

The paper provides an overview of current deepfake detection methods, demonstrating that these methods are flawed: they can be easily bypassed via white-box, black-box, and even agnostic attacks. This vulnerability arises because these methods suffer from domain shift, i.e., poor performance on out-of-domain samples.

**Strengths:**

**Comprehensive Evaluation:**

One strength of this work is its thorough evaluation. The authors test all three types of threat models (white-box, black-box, and agnostic) against various deepfake detectors.

**Human Evaluation:**

Additionally, it is valuable that the authors assess the quality of deepfake audio through human studies, showing that the attacks do not produce discernible changes to the audio.

**Weaknesses:**

**Vulnerability Already Well-Known:**

Deep learning-based audio models are already known to be susceptible to evasion attacks [1], with extensive literature on optimization-based and signal processing-based attacks. In fact, the signal processing family of attacks might be even better suited than optimization based ones (like PGD and iFGSM as used in the paper) since they do not even require query access to the model. At this stage, the community would benefit more from methods to enhance robustness in deepfake detection. To improve the paper, the authors could propose, implement, and evaluate new approaches to strengthen deepfake detection systems based on insights from this study.

**Lack of Information on Human Study:**

The authors do not provide enough details to assess their human study adequately:
- Did they obtain IRB approval before conducting the experiments?
- How many participants were involved in the study?
- How was the study structured?
- What was the null hypothesis?


Reference:

[1] The Faults in our ASRs: An Overview of Attacks against Automatic Speech Recognition and Speaker Identification Systems

**Questions:**

See above

**Details Of Ethics Concerns:**

Authors conducted a human study but do not provide any information about IRB approval.

---

### Official Review · Reviewer_TxjK · 2024-10-30

**Soundness:** 2
**Presentation:** 3
**Contribution:** 2
**Rating:** 3
**Confidence:** 4

**Summary:**

The paper asks the question if the audio deepfake (or deepfake speech) detection is reliable? To answer this question, it evaluates four deepfake detectors and uses different attack types specific to the nature of access to the model under evaluation. The paper draws insights from their experiments revealing that infact audio deepfake detectors are vulnerable and even simple attacks deteriorate their performance, making them highly unreliable. The analyses discussed in the work underscore the poor quality of current deepfake detectors.

**Strengths:**

- The problem of unreliable deepfake detectors is of primary concern to users, such as fact-checkers and investigators, who want to rely on their performance out-of-the-box. The paper hits at the core problem with deepfake detectors.
- The paper is overall clear to read and provides several results to ground their insights on.
- The presenting of takeaways and key findings makes the paper accessible for readers.

**Weaknesses:**

As this paper asks an evaluation question, **it needs to back it with substantive evaluations**. I suggest the following improvements:
- The complete evaluation is performed on 100 samples from 3 text-to-speech (TTS) datasets. Essentially basing all analysis on 300 samples. There are newer speech-to-speech datasets, such as DECRO [1]. Moreover, audio deepfake datasets are easy to create. If we have 100 speakers then we can create 100 * 99 (9,900) by inferring a speech generator, such as FREEVC, QuickVC, Suno-AI's Bark, TorToiSE, CoquiTTS. Subsequent filtering to retain high quality sample should be explored.
- If the datasets were in any form skewed, randomly sampling from them would carry forward those biases. A demographically and phonetically-diverse set of samples would give representation to practical edge cases.
- While white-box access to commercial deepfake detectors is harder to obtain, include them into black-box and model agnostic evals. Examples are resemble.ai, elevenlabs.io. In such case SimBA might not work, as confidence score is generally not given.

**A general lack of discussion**:The paper lists its insights but fails to make readers confident to believe what they are reading. For example, In 3.3, "ASSIST-L, being smaller, may form smoother decision boundaries, leading to better generalization and robustness against perturbations."  First, the language is hand-wavy as no evidence is given, and second, this insight itself is surprising.

[1] Transferring Audio Deepfake Detection Capability across Languages, WWW'23. https://github.com/petrichorwq/DECRO-dataset

**Questions:**

**Addition of the following information would help bolster reader confidence:**
- Description of how the human ratings were computed. Did you use a standard like P.808 for obtaining Mean Opinion Scores (MOS)? if not, why? Who were the raters?
- Description of the measure of the dataset being evaluated? Did the deepfake samples selected for evaluation fool humans? What was the baseline?
- Can authors defend their views in conclusion about Security by obscurity: "This underscores the critical need to restrict public access to SSD models and internal workings." What do the authors mean? Should the community not publish their works? How will anybody audit such systems then?

- Expand discussion of above discussed concepts by moving some of the plots to appendix.


**Stretch goal:**
- While limiting to a language is an understandable constraint, evaluating multi-lingual deepfakes might lead to novel insights.

---

### Official Review · Reviewer_K3R6 · 2024-11-05

**Soundness:** 2
**Presentation:** 3
**Contribution:** 2
**Rating:** 5
**Confidence:** 5

**Summary:**

The paper investigates the robustness of Synthetic Speech Detectors (SSDs) against deepfake audio created through modern Text-to-Speech (TTS) systems, focusing on malicious attacks intended to deceive these detectors. The study is the first to systematically evaluate SSD performance under adversarial attacks in three scenarios: white-box, black-box, and agnostic (transferability). The results demonstrate the vulnerability of SOTA SSDs, especially when attackers have even minimal access to the model, emphasizing the need for more robust detection solutions to combat evolving adversarial threats.

**Strengths:**

1. The paper covers a wide range of attack scenarios (white-box, black-box, and agnostic), providing a holistic view of SSD vulnerabilities under varying levels of attacker access.
2. The study employs both objective metrics (e.g., VisQOL scores) and subjective human ratings, ensuring that attack success considers not only detectability but also audio quality, suggesting a potential threat that currently, the perturbed audio can easily bypass the SSD without loss of audio quality.
3. Given the rapid advancements in TTS and voice cloning technology, this study highlights an urgent need for improved SSD security in the face of potential deepfake misuse.

**Weaknesses:**

1. The paper only evaluates the robustness of four SSDs against adversarial attacks, which may limit the overall scope and depth of its findings. More advanced SSDs should be incorporated.
2. The evaluation may not be sound. For example, in section 3.2, the paper claims that "Deepfake speech from TTS not seen during training is more likely to bypass SSDs.". However, it does not provide the benign detection performance of the SSDs across different datasets, which is crucial as existing SSDs often exhibit poor generalization across datasets. If the SSDs trained on ASVSpoof2019 already perform poorly on, for example, the In-the-Wild dataset, then the observed attack success rates cannot be directly attributed to the effectiveness of the attack itself. This also causes confusion in the transferability experiments, where the paper finds that some SSDs are extremely vulnerable on out-of-domain data.
3. Additionally, the human evaluation is insufficiently rigorous, and the experimental design lacks essential details. For instance, the paper does not specify the number of human subjects involved or the procedures used in conducting the experiments. These methodological details are important for assessing the validity of the human evaluation but are absent from the paper.
4. The study does not examine proprietary or commercially deployed SSDs, which could exhibit varying levels of robustness compared to open-source models. In addition, it would be also interesting to see if the adversarial audios crafted by open-sourced SSDs can transfer to commercially deployed SSDs.
5. While the paper effectively identifies vulnerabilities in SSDs, it does not provide a thorough examination of potential improvements or specific countermeasures that could help address these issues.
6. The insights presented in this paper are somewhat limited and largely familiar to the adversarial machine learning community.
7. Lack of details and code for reproducibility.

**Questions:**

Please refer to Weakness.

---

### Official Review · Reviewer_mtnx · 2024-11-05

**Soundness:** 2
**Presentation:** 3
**Contribution:** 1
**Rating:** 3
**Confidence:** 3

**Summary:**

This paper presents a series of experiments demonstrating the vulnerability of existing deepfake audio detection methods to adversarial attacks. Unfortunately, the quality of this submission does not meet the standards for ICLR.

**Strengths:**

- The manuscript is not too difficult to follow.

**Weaknesses:**

1.Lack of Novelty:
The challenges related to deepfake audio detection, particularly the decline in performance across domains, are well-established in the literature. The vulnerability of detection models to adversarial attacks is also widely recognized. Therefore, dedicating an entire paper to simply illustrating these issues does not contribute new insights to the field.

2.No Technical Contribution:
The paper primarily applies classic adversarial attack methods to examine the vulnerability of detection models, without offering any novel approaches or techniques.

3.Insufficient Detail on Model Training:
The paper lacks sufficient detail regarding the training process of the detection model. For example, was data augmentation employed to enhance the model’s generalization ability? There are various techniques that can improve model robustness, such as adding white noise to training data, which has been shown to increase resistance to adversarial attacks. The authors could investigate such approaches, perform a more comprehensive evaluation, and derive more valuable insights.

4.Poor Paper Organization:
The paper’s organization could be improved. For example, figures are often located far from the sections that describe them, making it difficult for readers to follow the flow of the paper.

**Questions:**

n/a

---

### Note · Authors · 2024-11-14

**Comment:**

We sincerely thank all reviewers for their constructive feedback. We plan to incorporate these insights to improve the work for future submissions.

**Withdrawal Confirmation:**

I have read and agree with the venue's withdrawal policy on behalf of myself and my co-authors.